# Visual Thinking Strategies as a Tool for Reducing Burnout and Improving Skills in Healthcare Workers: Results of a Randomized Controlled Study

**DOI:** 10.3390/jcm11247501

**Published:** 2022-12-18

**Authors:** Vincenza Ferrara, David Shaholli, Antonio Iovino, Sabrina Cavallino, Marina Andrea Colizzi, Carlo Della Rocca, Giuseppe La Torre

**Affiliations:** 1Art and Medical Humanities Lab, Pharmacy and Medicine Faculty, Sapienza University of Rome, 00185 Rome, Italy; 2Department of Public Health and Infectious Diseases, Sapienza University of Rome, 00185 Rome, Italy; 3Department of Medico-Surgical Sciences and Biotechnology, Polo Pontino, Sapienza University of Rome, 04100 Latina, Italy

**Keywords:** visual thinking strategies, burnout, skills, healthcare workers, randomized controlled study

## Abstract

**Objective**: The purpose of this study was to measure the effect that a learning method using art, such as Visual Thinking Strategies, can have on nursing students and residents in Hygiene and Preventive Medicine; we also aimed to improve skills of observation, communication, empathy and collaboration with the patient and other team members, and evaluate the impact on quality of life, burnout and positivity. **Methods**: The experimentation was based on administering a form (written assessment) before and after the intervention, to measure the impact of the method on improving some useful skills for the nursing and medical professions by inserting one image of an artistic type and the other of a clinical type, and asking participants to answer basic questions on the VTS method. Students participated in groups of eight in 90-min sessions for four meetings in the classroom and one at the museum, on a weekly basis. **Results**: The experimental study concerned a randomized controlled sample involving undergraduate nursing students who voluntarily participated in the survey. 84 students took part in the study, including 50 undergraduate nursing students (mean age 22.5, SD 2.7; 14 males and 34 females) and 34 residents in Hygiene and Preventive Medicine (mean age 28.7, SD 2.6; 8 males and 26 females). For the artistic image, the median of the total score for all skills was −1.5 for the control group and 3 for the intervention group (*p* = 0.002), registering an increase for the Delta identified items, which was −1.5 for the control group and 1 for the intervention group (*p* = 0.007). For the clinical image, the median of the total score for all skills was 0 for the control group and 2 for the intervention group (*p* = 0.025), recording an increase for Delta observation skills, found to be 0 for the control group and 1 for the intervention group (*p* = 0.007). **Conclusions**: Most students who participated in the intervention reported considerable improvements in the skills the method promises to improve, and a reduction in personal and work burnout scores, suggesting that the use of the VTS method in health professions curricula is viable.

## 1. Introduction

Possession of the formal qualification of nurse, in charge of general nursing, must prove that the professional in question is capable of applying the following skills: diagnosing independently, working effectively with other health care professionals, facilitating people to acquire healthy lifestyles and self-care skills, initiating immediate interventions independently, counseling, educating and supporting individuals and their families, ensuring and evaluating the quality of nursing care independently, communicating comprehensively and professionally, and verifying the quality of care provided [1,2]. The core competencies for good nursing practice, however, need a well-defined chronological order of action called the “Nursing Process”; this includes a series of planned steps and actions aimed at meeting the needs and solving the problems of the person being cared for and their family members.

The nursing process functions as a systematic guide to patient-centred care, with 5 sequential steps. These are assessment, diagnosis, planning, implementation, and evaluation. For each step, it is possible to individuate an important skill. Assessment is the first step and involves critical thinking skills and data collection, subjective and objective. The formulation of a nursing diagnosis requires good observation and problem-solving skills. A good capacity for communication is a very important skill for planning activities [2].

In the context of skill development and learning development studies, several strategies can be implemented that are useful in achieving these goals; these can adopt other disciplines, such as the humanities [3], and strengthen the Medical Humanities approach to basic and continued education.

Art can be used to develop observational skills in students, and can provide an innovative mechanism for students to develop higher levels of awareness and sensitivity to change [4].

Art has been used to improve observational, critical thinking, communication and other skills in a variety of ways in nursing education [5]. The observation of art can help to develop visual skills and is used to manage and reduce stress, ultimately influencing an approach with patients [6].

Several studies have investigated how to teach and improve skills such as visual literacy, teamwork, critical thinking, empathy and active listening among healthcare students. One of the techniques used by art-based programs is the Visual Thinking Strategies (VTS). It is a pedagogical approach and an evidence-based method that uses observation and discussion on art [4].

In this study, visual art was used, specifically the Visual Thinking Strategies (VTS_ method, as a tool for learning. With the VTS method, the artwork becomes a tool for the development of certain skills [7,8,9,10]. This method is based on observation and then interpretation of the artwork, which is followed by a discussion among the participants. This process is guided by a facilitator, who encourages participants to bring out and use their observation and reflection skills, but without being influenced in any way.

This method can be implemented within classrooms, museums, or via distance learning; this is carried out through three questions that the facilitator asks the participants about an image, and the group is invited to observe:-What is going on in this picture?-What do you see that makes you say that?-What more can we find?

Through these questions, a constructive discussion can take place as each participant contributes to the identification of the story depicted. Important is the choice of images, which, rich in details and easily resolved ambiguities, must correspond to the cultural level of the group and be unfamiliar to the participants, precisely because, from these, a discussion must be activated based on their knowledge and skills.

Using visual art in the training of health professionals corresponds to three different levels of action and intervention: the first is related to the development of a particular thinking capacity that knows how to be flexible and skillful in linking different fields or sectors; the second refers to the development of professional identity through reflection on aspects related to the relationship with one’s patients; and the third is related to the acquisition of teamwork skills since these are an essential aspect in the development of relational skills and the patient-centered set-up; this constitutes a space not only for recognizing and sharing one’s mental states, but legitimizes the relevance that these aspects have in the professional field. The activities of analysis, confrontation and discussion, carried out through art, enable the student to improve his or her critical thinking and problem-solving skills, to develop greater empathy towards the patient and more respect towards others, whether they are patients or colleagues [11,12,13,14].

In this trial, it was decided that third-year nursing students would be the intervention group using VTS, to provide additional useful data that can help improve nursing education. The aim of the study was to assess whether Visual Thinking Strategies can be useful tools for reducing burnout and improving skills in healthcare workers.

## 2. Materials and Methods

The project was proposed to the third year of the Z course of Nursing in Pomezia of “Sapienza University of Rome” 2018/2019, and to residents of Hygiene and Preventive Medicine in the academic year 2019/2020 and 2020/2021.

The randomization process was carried out as follows. A complete list of students and residents was obtained. The allocation of participants to the intervention or to the control groups was realized using a list of random numbers generated with EpiCalc-2000, separately for females and males, as well as for students and for residents. Concealment of the sequence was put in place until assignment occurred. Since the control group was assigned to no intervention, blinding was not feasible. The participants signed an informed consent before starting the trial.

In total, 84 students took part in the study, including 50 undergraduate nursing students (mean age 22.5, SD 2.7; 14 males and 34 females) and 34 residents in Hygiene and Preventive Medicine (mean age 28.7, SD 2.6; 8 males and 26 females) (Figure 1).

The project included five meetings, four at university classrooms, and a final meeting at the “National Etruscan Museum” in Villa Giulia and at the National Gallery in Rome. In order to measure the impact of the method on the improvement of certain skills, a form (written assessment) was administered; this form included the insertion of two images, one of an artistic type and the other of a clinical type, asking participants to answer the basic questions of the VTS method: “What is happening in the image? What are the visual elements that support your hypothesis?” (Figure 2).

This form was administered before the first meeting and after the VTS activities of last meeting at the museum. The VTSkill Rubric was used to assess the impact of VTS activities. The writing rubric VTSkill used was designed to assess critical thinking, observation and attention skill, linguistic expression for communication capacity, and problem solving [15].

The Analytic Rubric VTSkill for learning outcome assessment features a grid of “competences” (dimensions on the rows) and “levels” of achievement (score on the columns). The rater, usually the instructor of the course, evaluates the participant performance in each dimension and assigns values corresponding to a specific set of criteria, derived from the specific literature on the topic. Specifically, the VTSkill grid allows the analysis of the abilities of critical thinking, the ability to observe and pay attention, language expression, the number of words used, problem solving, and elements identified. These abilities were measurement reading the written assessment and comparing the results before and after VTS activities.

Students participated in groups of eight in 90-min sessions for four meetings in the classroom and one at the museum on a weekly basis. The number of participants in at least four meetings was 25 students. A format already tested with medical students was used [12].

During the first lecture, the theme of the relationship that has existed for centuries between art, medicine and healing was introduced, the VTS method was explained, how and why it can be exploited, and through the observation of the work “The Doctor’s Visit” by Jan Steen, 1661–1662, Victoria and Albert Museum, London (Figure 3), an initial classroom experience was made.

This strategy provides the tools for observing and analyzing an image and can be compared to the activities that physicians and nurses perform during clinical practice. The patient, first examined individually, is then observed by each member of the team and each of them expresses his or her opinion with the expert mentor playing the role of facilitator in leading the whole team toward the formulation of a shared and accepted hypothesis.

As in the first meeting, each session included a VTS discussion on an image of a work, an individual written VTS practice with a subsequent collective discussion, and an active listening practice with the drawing of an image described by a facilitator. The chosen images also sought to address the need for resolving ambiguities and correlation with the care environment. Indeed, this was the case of “Nosocomio”, by Silvio Giulio Rotta, 1895, The National Gallery in Rome (Figure 4), where the situation in the ancient asylums is depicted, thus combining both practice for skill development and knowledge related to one’s profession.

All lectures were aimed at preparing the student for the last meeting held in the “National Etruscan Museum” of Villa Giulia in Rome for nursing students, and at the National Gallery in Rome for Hygiene and Preventive Medicine residents. Two workshop activities were held during the meeting in the museum space, one divided into groups and the other individual. In the first workshop, a facilitator followed the group through the VTS method on a work, while in the second part they went on to carry out an individual questionnaire, always related to the VTS questions to be filled out individually in front of a work.

A satisfaction survey was administered to the participants to ask for their evaluation of the proposed activities, using a Numerical Rating Scale (NRS) score.

### Statistical Analysis

For the analysis, we focused on skills that VTS can help improve.

Relative to clinical variables, the following were assessed:-Health-related quality of life, using the SF12 questionnaire, which enabled the calculation of Mental Composite Score (MCS) and Physical Composite Score (PCS).-Positivity, using Caprara’s Positivity scale.-Burnout, using the Copenhagen Burnout Inventory, which allowed the following scores to be calculated: Personal burnout, Work burnout, Client burnout.

Statistical evaluation was performed using Microsoft Excel spreadsheet software and Statistical Package for Social Sciences software (SPSS 26.0, Chicago, IL, USA). Quantitative variables were expressed as mean ± standard deviation, qualitative variables as a percentage.

Between-group differences were assessed by employing the Mann–Whitney test for between-group comparison, and the Wilcoxon test for paired samples for within-group comparison. The significance level was set at *p* < 0.05.

## 3. Results

In Table 1, a description of the demographic characteristics of the participants is given by intervention and control group.

The data were divided by the two types of images offered during the intervention, namely artistic and clinical.

Regarding artistic images (Table 2), we immediately notice important differences between those who did not participate in the intervention and those who participated in the activities:-observational skills: the median differs to be 0 for the control group and 1 for the intervention group, with a maximum of 4;-language expression: where for the control group the median is 0, for the intervention group it is 1, with a maximum of 4;-problem solving: similar to the other two, the control group is 0, while the intervention group maximum was 1.

Thus, the total artistic image score presents a median for the control group that is −1.5, while for the intervention group it rises to 3.

For the clinical image (Table 3), the data also prove to be important in terms of treatment outcome:

Here, too, we record considerable improvements:-observational skills: 0 for the control group and 1 for the intervention group with a maximum of 3;-language expression: 0 for the control group and 1 for the intervention group with a maximum of 3;-problem solving: maintains the same median of 0, however, has a maximum 3 for the intervention group.

Again, the total clinical image score presents a median of 0 for the control group, and a median of 2 for the intervention group.

Table 4 shows the results for health-related quality of life, positivity and burnout.

It can be observed that, in the intervention group, there is an increase in MCS scores, and a decrease in the score on personal burnout. As for the control group, however, there is an increase in all three burnout scores, and a decrease in the PCS score. Between the intervention and control groups, there are no differences for the scores analyzed at time 0, while significant differences are observed for the personal (*p* = 0.040) and work (*p* = 0.016) burnout scores, with the lowest scores are in the intervention group.

Finally, a NRS score of 8.3 (SD 1.2) was registered for the evaluation of the proposed activities.

## 4. Discussion

This study sought to test the extent to which the use of a learning method using art can improve skills useful to the medical and nursing professions. Data analysis shows considerable increases in the improvement of skills such as observation, language expression, and problem solving for both art imagery (total score: control group = −1.5; intervention group = 3) and clinical imagery (total score: control group = 0; intervention group = 2). A parallel study analyzed the impact of VTS on stress and burnout, showing positive results.

An important finding that should not be underestimated is how interesting and positive the students found the course. This interest could be an expression of a real need to improve their skills, not only from a content/theoretical point of view, but also in relation to the approach to the lesson, putting the student at the center of a group that has a common purpose: patient care.

The results obtained are perfectly in line with the qualitative studies referring to the application of the Visual Thinking Strategies method that the literature provides us with, in regard to the improvement of health professionals [16,17,18,19]. It seems that the participants of the various similar studies, after experiencing VTS, are able to evaluate their patients more critically and glimpse more nuances and details. Moreover, after experiencing VTS, they realized that they were more accurate in providing key information to other members of the health care team, in the clinical setting. Another commonality, among the various studies, is the development of skills in emotional recognition, cultivation of empathy, identification of stories and narratives typical of NBM (Narrative-Based Medicine), and awareness of multiple perspectives; these are all characteristics that help to see the patient in a more complete and comprehensive picture.

In addition, most students rated the use of this exercise as an enjoyable learning experience.

This VTS learning experience exposed vulnerabilities in participants’ learning experiences, and helped uncover ways in which students felt safe to explore and expand their learning. The VTS experience with art also revealed a way for students to think and see differently through the careful consideration of thinking and seeing others.

The use of the randomized controlled trial as a research method ensures a rigorous approach by ruling out the possibility that there may be, for example, a bias in the sample toward those individuals who are more interested in art; these individuals, in theory, could skew the results due to greater observational skills.

It would be interesting to develop, in the future, studies with an even larger number of health professionals, also involving other disciplines such as physiotherapists, social workers, etc., and to increase the number of encounters, using the VTS method, in order to have even more complete results based on the understandable individual variations in learning development and different communicative and observational skills.

## 5. Conclusions

The literature referring to the studies, cited above, on the impact of the use of art in this field, and the results, presented both qualitatively and quantitatively, demonstrate an important effectiveness of the VTS method; this is evident in both the improvement of skills useful to the health professions, and a reduction in burnout levels and an increase in mental score.

## Figures and Tables

**Figure 1 jcm-11-07501-f001:**
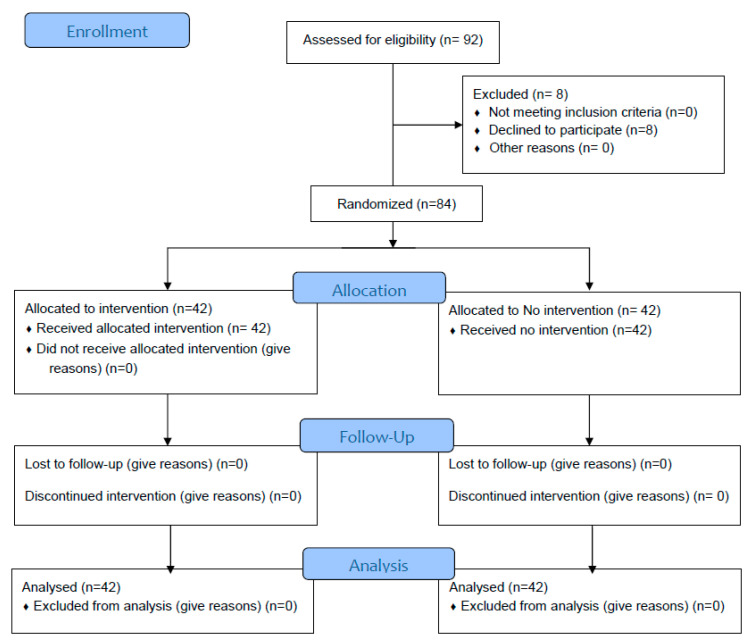
CONSORT 2010 Flow Diagram.

**Figure 2 jcm-11-07501-f002:**
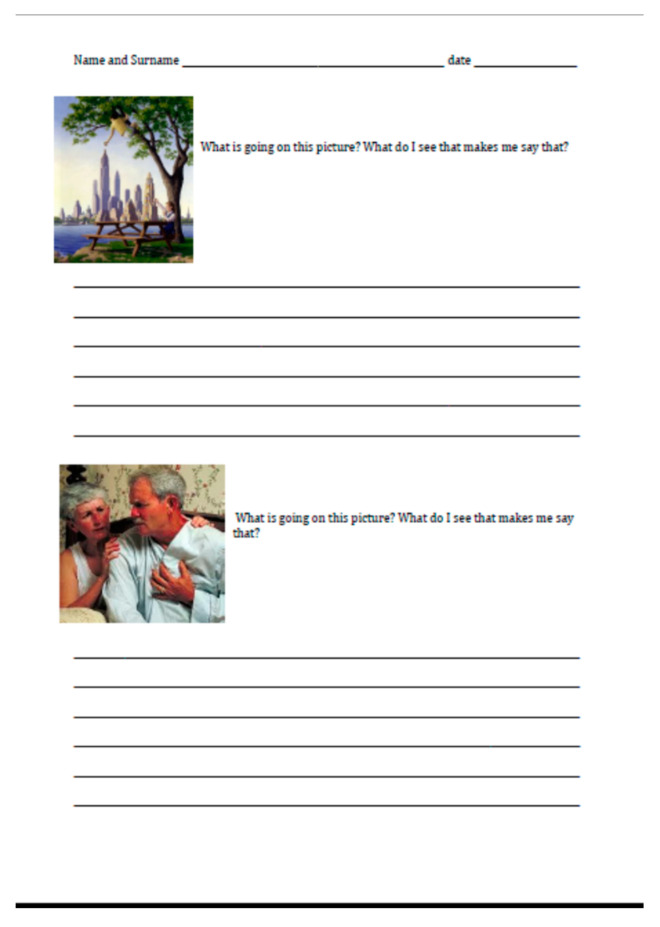
Write assessment of VTS activities.

**Figure 3 jcm-11-07501-f003:**
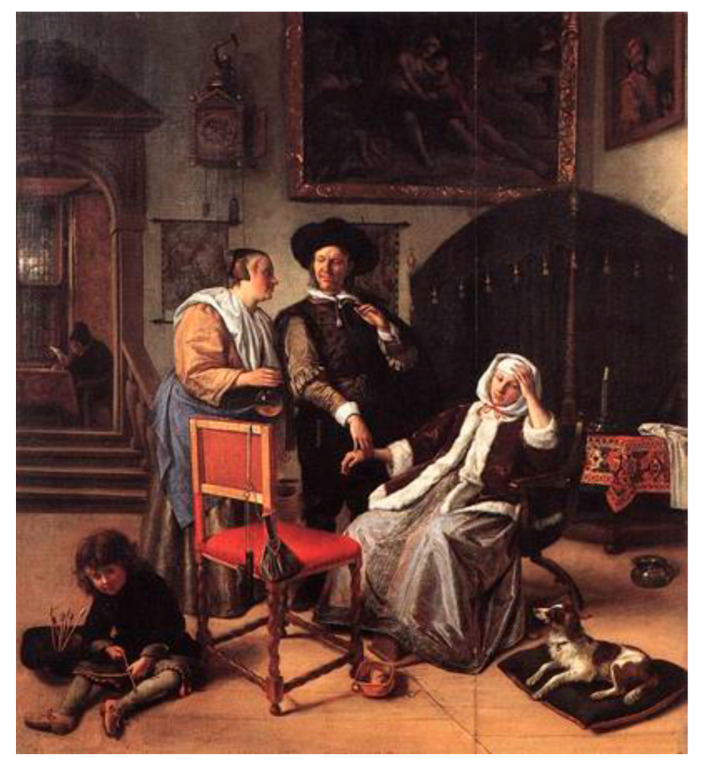
“The Doctor’s Visit” by Jan Steen, 1661–1662, Victoria and Albert Museum, London.

**Figure 4 jcm-11-07501-f004:**
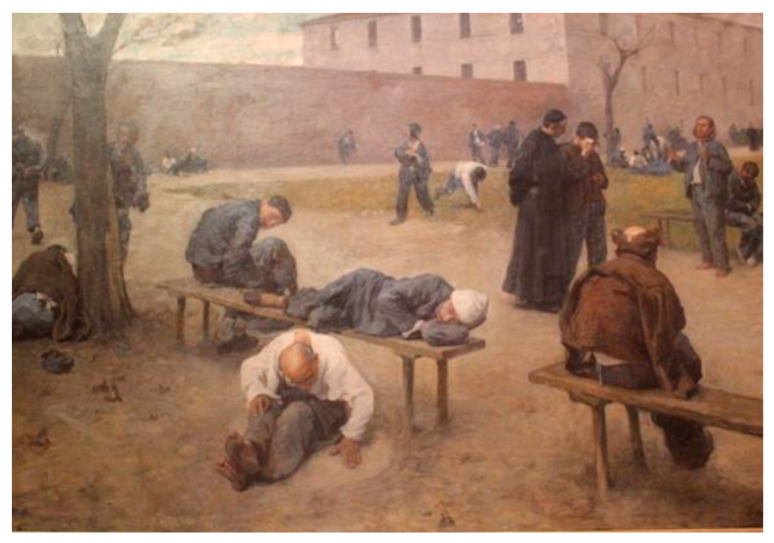
“Nosocomio,” Silvio Giulio Rotta, oil on canvas (200 × 300cm) 1895, National Gallery of Modern Art in Rome.

**Table 1 jcm-11-07501-t001:** Demographic characteristics of the participants.

	Control Group Number, Mean (SD)	Intervention Group Number, Mean (SD)
Total Number	42	42
Nurses students	25	25
Residents	17	17
Age	25.1 (2.7)	25.0 (2.6)
Gender		
Females	30	30
Males	12	12

**Table 2 jcm-11-07501-t002:** Results related to artistic image.

	Control Group Median (Min–Max)	Intervention Group Median (Min–Max)	*p*
Delta Critical thinking	0 (−1–1)	0 (−2–2)	**0.044**
Delta Observation skills Clinical Attention	0 (−1–1)	1 (−1–4)	**0.006**
Delta Linguistic expression	0 (−1–1)	1 (−3–4)	**0.009**
Delta Word count	−11 (−20–6)	6 (−43–48)	0.081
Delta Problem solving	0 (−1–1)	1 (−2–4)	**0.008**
Delta Identified elements	−1.5 (−5–2)	1 (−4–7)	**0.007**
Total score artistic image	−1.5 (−4–4)	3 (−6–14)	**0.002**

**Table 3 jcm-11-07501-t003:** Results related to clinical image.

	Control Group Median (Min–Max)	Intervention Group Median (Min–Max)	*p*
Delta Critical thinking	0 (−1–1)	1 (−1–3)	0.162
Delta Observation skills Clinical Attention	0 (−1–1)	1 (−1–4)	**0.007**
Delta Linguistic expression	0 (−1–2)	1 (−1–3)	**0.028**
Delta Word count	0.5 (−23–8)	9 (−37–64)	0.081
Delta Problem solving	0 (−1–1)	0 (−1–3)	**0.028**
Delta Identified elements	−0.5 (−1–3)	1 (−2–7)	0.107
Total score clinical image	0 (−4–2)	2 (−2–12)	**0.025**

**Table 4 jcm-11-07501-t004:** Results related to MCS, PCS, positivity score and burnout.

	Control Group	Internvention Group	*p*
PCS12_pre	50.91	49.79	0.672
PCS12_post	44.64	48.46	0.108
*p*	**<0.001**	0.214	
MCS12_pre	43.01	37.78	0.220
MCS12_post	40.72	43.05	0.899
*p*	0.08	**<0.001**	
Positivity_pre	3.31	3.31	0.882
Positivity_post	3.31	3.25	0.783
*p*	1.00	0.925	
Personal_burnout_PRE	50.00	50.00	0.317
Personal_burnout_POST	58.33	45.83	**0.040**
*p*	**<0.001**	**<0.001**	
Work_burnout_PRE	30.36	35.71	0.317
Work_burnout_POST	46.43	35.71	**0.016**
*p*	**<0.001**	1.00	
Client_burnout_PRE	18.75	29.17	0.120
Client_burnout_POST	20.83	29.17	0.983
*p*	**<0.001**	1.00	

## Data Availability

Data are available upon request to the authors.

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
