# Peer review of "Visual Thinking Strategies as a Tool for Reducing Burnout and Improving Skills in Healthcare Workers: Results of a Randomized Controlled Study"

_jcm, 2022, doi:10.3390/jcm11247501_

Round 1
Reviewer 1 Report
The study investigates the use of Visual Thinking Strategies (VTS) as a tool for reducing burnout and improving skills in healthcare workers. The research paper covers an interesting topic and provides an interesting take on the use of VTS to reduce burnout in nursing students whilst improving specific sets of skills which are essential in clinical practice.
Overall, this research paper is well-written and understandable, however, in different places, it feels rushed. More in detail, different notions and concepts are often implicit and in some places missing, thus leaving a naïve reader wondering about the reasons for specific choices. These points will be highlighted in the list below.
Furthermore, results should be reported according to standards. All the test between and within groups are missing their respective test statistics (U and W) and effect size. These are important to provide a correct interpretation of the results.
By taking into account the concerns listed below, I would suggest reconsidering this article after major revision.
Major concerns:
Most of the major concerns listed below focus on the need for further details and clarity:
- Line 49: “The authors state that the “Nursing Process” includes a series of planned steps and actions aimed at meeting……” It would be nice to see here a discussion of these steps and actions as it would be useful to understand how these relate to the skills listed in the previous lines
- Line 51: Similarly to the previous comment, an example of these strategies would be useful here. Do they have an impact on one step, or on multiple?
- Line 56: Is there a reason why VTS was chosen over other methods that could have provided similar results?
- Line 95: A form used to assess the impact of the method on the improvement of “certain” skills is introduced here. This form was provided before and after VTS. However, it is not clear how. How were the responses provided in this form assessed? What kind of process did the information go through?
- Line 147 to line 161: The list of skills that VTS can improve is provided here. However, these should be listed earlier in the article. Also, it would be useful to explain more in-depth how and why VTS can improve these specific sets of variables. Furthermore, it is not clear how all these skills were measured. Much more detail is needed here.
- Line 174: sentence “we take the median as the reference value is set p = 0.05”. While it is correctly stated before that the median is chosen because data are not normally distributed, it is not clear its relationship with the p-value in this sentence.
- Line 177 and all Tables reporting statistical analysis. These are missing important pieces of information.
Minor concerns:
- Line 172: Participants’ info is placed into results. Furthermore, the flow chart is placed after the first result table.
- In two occurrences in the discussion section is stated that students provide positive feedback towards the activities. Whilst it is not the focus of the paper, it would be interesting to explain how this notion was obtained. On line 238 is stated that this is an important finding, however, it is not clear how this was obtained and/or whether any type of analysis was undergone to obtain this finding.
Reviewer 2 Report
In this study, the authors conducted a randomized controlled trial to examine the impact of the Visual Thinking Strategies on reducing burnout, improving skills and quality of life among healthcare workers in Italy. There are many areas that need to be improved and clarified, and the overall structure of this manuscript should be improved too. Below are my major comments.
1. Please describe how this randomized controlled trial was conducted. What is the assignment mechanism? Concealment? Blinding? And has this randomized trial been registered formally? These are the key for this study.
2. The authors missed the Table 1 to describe the demographic characteristics by intervention and control group.
